# Fake news, real war

Jais Adam-Troian
Misinformation is often framed as a cognitive failure, focusing on the vulnerabilities of those who believe it. But misinformation often stems from deliberate disinformation campaigns—which should be considered proactive intergroup aggression. This shift in perspective moves the focus from targets to actors, calling for interventions directed at those who create and spread falsehoods.

The rapid growth of misinformation research has deepened our understanding of how false narratives propagate and shape public beliefs. Much of this work is rooted in what can be termed the "cognitive paradigm," which conceptualizes misinformation primarily through its interaction with individual cognitive processes. Within this framework, research focuses on how cognitive biases, heuristics, and other psychological mechanisms facilitate belief in falsehoods, as well as how misinformation narratives exploit these vulnerabilities to achieve widespread dissemination[1].

The cognitive paradigm relies on the distinction between misinformation and disinformation, with misinformation generally defined as false information spread without intent to deceive, while disinformation is understood as deliberately deceptive. By attributing the issue of human agency to disinformation, this distinction has enabled researchers to conceptualize misinformation narratives as cognitive contaminants—comparable to viral infections that spread from person to person[2]. As a result, the field has concentrated on identifying cognitive risk factors for misinformation belief, such as motivated reasoning, alongside the psychological and social mechanisms that facilitate its propagation through networks. This focus, grounded in the assumption that misinformation spreads primarily due to individual cognitive vulnerabilities, has driven significant progress in the development of interventions aimed at enhancing cognitive resilience[3].

## Misinformation as relational aggression

Yet, I suggest that misinformation often stems from deliberate acts of proactive intergroup aggression, in which false narratives and rumors are actively created and disseminated to harm outgroups while benefiting ingroups. This framing positions misinformation as the outcome of an intentional behavior embedded within intergroup dynamics: disinformation behavior. I thus propose that much misinformation can be more accurately understood through the lens of proactive relational aggression, a form of aggression that seeks to damage others through social manipulation, rather than through direct physical violence[4].

While the distinction between misinformation and disinformation is useful, it remains problematic if misinformation is simply viewed as false information. In this case, misinformation research becomes indistinguishable from the broader study of belief, overlapping substantially with core areas of psychology. However, a closer look at the field reveals a concentrated focus on specific types of false information—particularly fake news related to politics, science, and public health (e.g., infodemics), which is initially crafted with the intent to deceive. My argument does not deny the importance of research focusing on recipients or on misinformation that is not the result of malicious actors. Rather, it highlights the need to shift attention toward the intentional actors behind disinformation campaigns. The most consequential misinformation beliefs—such as narratives about the legitimacy of the invasion of Ukraine or the "stolen" 2020 U.S. election—are not simple byproducts of cognitive biases. Rather, they are the result of systematically orchestrated operations, carefully designed and disseminated by politically motivated entities to shape public opinion[5]. In essence, much of what is labeled as misinformation is, in fact, disinformation.

Going back to the epidemiological metaphor, these narratives are not random "viruses" spreading from person to person as a result of their vulnerabilities, but rather engineered bioweapons, deliberately introduced into the information ecosystem. The field has made progress by focusing on the targets of disinformation, which has led to the development of cost-effective cognitive "vaccines"—such as critical thinking or inoculation interventions. To develop more powerful treatments, it needs to start addressing the actors responsible for producing and disseminating disinformation, who constitute a critical part of the problem. In other words, mitigating public susceptibility is necessary, but insufficient without confronting the architects and amplifiers of disinformation— the true pathogenic source.

## A strategically deployed weapon

Unlike reactive aggression, which is impulsive, emotionally driven, and often a response to perceived threats, proactive aggression is calculated, strategic, and goal-oriented[6]. It is designed to weaken, isolate, or undermine its targets while simultaneously enhancing the status, cohesion, or power of the aggressor group. Disinformation achieves this by eroding trust within and between communities, sowing division, delegitimizing institutions, and amplifying societal fractures. Whether deployed by state actors in geopolitical conflicts, political operatives during elections, or interest groups seeking to manipulate public opinion, disinformation operates as a weapon of psychological and social warfare—engineered not just to misinform, but to destabilize, demoralize, and divide. Applying this framework to misinformation clarifies its function in intergroup conflict. Much like relational aggression at the individual level—where gossip, exclusion, and reputation attacks serve as tools for dominance—disinformation operates as a collective-level weapon aimed at fracturing adversaries.

Disinformation is the intergroup equivalent of malicious gossip[7], functioning to damage rival coalitions while bolstering one's own. The difference lies in the scale and number of third parties. Unlike gossip among small real-life networks, misinformation makes use of public opinions to reach their target (e.g., rival company, politician or government). This perspective highlights why debunking alone is not optimal: falsehoods are often socially useful to the groups who spread them.

Just as some individuals use gossip to solidify alliances or undermine rivals, groups deploy disinformation for strategic advantage. Thus, interventions focused solely on changing individual minds without addressing the strategic value of disinformation for those who spread it are

fundamentally incomplete. In fact, research shows how few individuals share fake news because it generally damages their reputation[8]. Spreaders are thus those who see reputational benefits in it. In terms of application, this means that name-and-shame type of responses could be viable instruments in the toolbox of interventions to deter disinformation behavior[9].

## A research and policy agenda for misinformation-as-aggression

The overreliance on interventions rooted in cognitive assumptions has resulted in strategies—such as inoculation or accuracy nudging—that often yield modest effect sizes, if any[10]. More critically, this approach frames the problem as residing primarily within the cognitive vulnerabilities of individuals targeted by disinformation, effectively placing the burden on the victims of aggression while neglecting the behaviors of the aggressors—those who create and disseminate disinformation. To address this gap, behavioral science must adopt a broader research agenda focused on understanding the drivers of disinformation behavior and identifying effective interventions to reduce its prevalence[11]. Building a robust framework for countering misinformation requires attention to three key priorities: classifying and categorizing disinformation tactics, developing predictive models of disinformation behaviors, and designing systemic interventions that target the sources of misinformation narratives.

Just as aggression research has established taxonomies of aggressive behaviors, misinformation research must advance toward a structured classification of disinformation types. This would enable researchers to determine whether different tactics—such as social media campaigns, hybrid efforts, and lobbying operations—share common psychological underpinnings or vary across cultural contexts. There is a critical need to shift focus toward understanding the psychological, social, and economic motivations of disinformation creators, including state actors, political operatives, and individual influencers. Insights from aggression research suggest that intentional harmful actions have identifiable psychological predictors, and similar models should be developed to elucidate the mechanisms underlying disinformation behavior.

Applying the Misinformation-as-Aggression paradigm necessitates a conceptual shift from individual-level interventions to systemic approaches. For example, recent research focusing on corporate misinformation indicates that legal consequences—such as imposing severe penalties for spreading misinformation online—can indeed have deterrent effects[12]. Policymakers could extend legal accountability to disinformation architects, particularly those involved in coordinated campaigns. Additionally, insights from game-theoretical models suggest that the most effective strategy to enforce cooperation is through systematic, proportionate retaliatory measures against uncooperative actors (i.e., variants of tit-for-tat strategies)[13]. Targeting such actors could be cost-effective if they are influent, as it can lead to trickle-down effects akin to targeting entire networks (i.e., 100% of individuals) through social contagion induction[14]. Accordingly, researchers should assess the effectiveness of proactive state penalties against malevolent actors, including sanctions, diplomatic responses, and the public denunciation of malign influence operations by government officials.

Finally, if disinformation functions as malicious intergroup gossip, then strategic counter-gossip should be explored as a potential deterrent[15]. Liberal democracies do not need to manufacture falsehoods; instead, they can amplify truthful narratives about the corruption, human rights abuses, and failings of authoritarian regimes or actors that rely on disinformation. While ethical concerns surrounding counter-disinformation campaigns must be carefully considered, transparency and the factual reporting of adversary misconduct remain legitimate and effective tools for deterrence.

As in other domains of conflict, passivity invites aggression. Disinformation is not only a problem of belief—it is a problem of power. A comprehensive response must integrate behavioral science insights with legal accountability and strategic deterrence, targeting both the cognitive vulnerabilities of individuals and the behaviors of disinformation architects. By broadening the scope in this way, we can build a more effective and resilient information ecosystem.

**Jais Adam-Troian** (iD) ✉
Heriot-Watt University Dubai, Dubai, United Arab Emirates.
✉e-mail: j.adam-troian@hw.ac.uk

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

## Competing interests

The author declares no competing interests.
