## [Transparent Peer Review file · Communications Psychology]

Fake News, Real War: Reconceptualizing Misinformation as Aggressive Behavior.

Corresponding Author: Dr Jais Adam-Troian

Version 0:

Decision Letter:

Dear Dr Adam-Troian,

Thank you for your patience during the peer-review process. Your manuscript titled "Fake News, Real War: Reconceptualizing Misinformation as Aggressive Behavior." has now been seen by 2 reviewers, and I include their comments at the end of this message.

The reviewers are in principle enthusiastic about your work. However, they also mention a number of concerns. We are interested in the possibility of publishing your manuscript in *Communications Psychology*, but would require substantive revisions before we make a decision on publication.

The present version of this Comment includes a strong and important call to action: focusing on the sources, or perpetrators of misinformation (and considering the spread of misinformation as relational aggression). As you lay out in the beginning and later again, this is conceptually equivalent to focusing on the part of misinformation that is the consequence of disinformation, or put differently, focusing on the perpetrators and mechanisms of disinformation.

To make this argument, it is not necessary to deny the contributions that misinformation research (the "cognitive paradigm") has made, nor to argue that all misinformation is the result of deliberate nefarious action (see Reviewer #2). The way the piece is currently written, it needlessly antagonizes readers who will feel that misinformation research that focuses on the recipients is given short shrift and that your proposal offers as an antithesis a similarly reductionist view when a synthesis of both streams would move the field much farther forward.

In other words, misinformation research has offered insights into what makes people vulnerable to misinformation and what protects from misinformation spread regardless of whether it's maliciously spread misinformation (i.e. outcome of disinformation) or not. As you explain, it is equally or even more important to focus on actors and on the type of misinformation that is the outcome of disinformation, but that's not to invalidate the existing science and it can be achieved within the nomenclature (misinformation vs disinformation) that you confirm to be largely useful. Although controversy and what Reviewer #1 calls "pithy" arguments can contribute to the success of a piece, it is equally important to get readers on board and avoid statements that they would consider strawman arguments regarding past contributions.

To aid you with that task of revising the work, I have included a marked-up version of your manuscript.

* **TRANSPARENT PEER REVIEW:** *Communications Psychology* uses a transparent peer review system. This means that we publish the editorial decision letters including Reviewers' comments to the authors and the author rebuttal letters online as a supplementary peer review file. We publish these records for all accepted manuscripts. However, on author request, confidential information and data can be removed from the published reviewer reports and rebuttal letters prior to publication. If your manuscript has been previously reviewed at another journal, those Reviewers' comments would not form part of the published peer review file.

If you have any questions about any of our policies or formatting, please don't hesitate to contact me.

Please use the following link to submit your revised manuscript and a point-by-point response to the referees' comments (which should be in a separate document to any cover letter):

Link Redacted

We hope to receive your revised paper within 4 weeks; please let us know if you aren't able to submit it within this time so that we can discuss how best to proceed. If we don't hear from you, and the revision process takes significantly longer, we may close your file.

Please do not hesitate to contact me if you have any questions or would like to discuss these revisions further. We look forward to seeing the revised manuscript and thank you for the opportunity to review your work.

Best regards,
Marika Schiffer, on behalf of

Jennifer Bellingtier, PhD
Senior Editor
Communications Psychology

REVIEWERS' COMMENTS:

Reviewer #1 (Remarks to the Author):

Review of MS COMMSPSYCHOL-25-0096

by Adam-Troian

Reviewer: Stephan Lewandowsky

Summary and Overall Recommendation

This commentary argues that misinformation is frequently actually disinformation (i.e., created and spread with intent), and that therefore the conventional research focus on recipients (what the author calls the "cognitive paradigm") is misplaced. Instead, focus should be on the actors that disseminate disinformation with the intent to cause harm. On this view, misinformation is best understood as "a form of aggression that seeks to damage others through social manipulation", which renders interventions that focus on individual minds "fundamentally incomplete." An alternative research agenda should (1) classify disinformation tactics, (2) develop predictive models of disinformation behaviors, and (3) design systemic interventions.

Turning to evaluation, I am in two minds about this paper: On the one hand, I agree with almost everything that is said, and I found the pithy and straightforward argumentation quite refreshing. On the other hand, I am not sure whether the "bite size" of this paper is appropriate for the journal. This reads more like a brief opinion piece than an academic article that is extensively supported by citation and evidence. The invitation to review unfortunately did not indicate what type of paper this is, but I checked the submission types (<https://www.nature.com/commsspsychol/submit/content-types#comment>) and if this is intended as a Comment, then I would definitely support publication.

Detailed comments
[line#]

16 A citation in support of this distinction might be helpful here.

114 According to the journal, "Readers are alerted that the reliability of data and conclusions presented in this manuscript is currently in question. Appropriate editorial action will be taken once this matter is resolved." This may not be the best source to cite until this issue is resolved?

Reviewer #2 (Remarks to the Author):

In the manuscript "Fake News, Real War: Reconceptualizing Misinformation as Aggressive Behavior." The authors argue that there is a need to reconceptualize our understanding of "misinformation" and that an aggressive behavior perspective offers a solution. After reading the arguments I remain unconvinced. I explain my concerns in more detail below.

At several occasions the authors argue that "Misinformation (...) is the downstream effect of deliberate disinformation campaigns." Or that "Misinformation stems from deliberate acts of proactive intergroup aggression," or that "Misinformation can be defined as the successful outcome of disinformation behaviors." With this framing the authors exclude false

information that is the result of errors from the definition of misinformation. I can only think of two settings why we, as a community, should deviate from earlier definitions and follow that reasoning (e.g.): a) the old definitions are too broad b) the old definitions are too narrow.

a) Common definitions of misinformation (e.g., Vraga & Bode, 2020) do include disinformation and thus also deliberate disinformation campaigns and deliberate acts of spreading false or misleading content. Following this, I do not see how this situation causes the need for reconceptualization here and the authors also introduce the known difference between misinfo and disinfo. That being said, it could be interesting to study the effectiveness of interventions such as sanctions against malevolent actors but this interesting research question does not imply the need to redefine a concept that allows to do just that, that is, study interventions against disinformation.

Vraga, E. K., & Bode, L. (2020). Defining misinformation and understanding its bounded nature: Using expertise and evidence for describing misinformation. *Political Communication*, 37(1), 136-144.

b) That leaves us with the recent definitions being potentially too broad. In line with this, the authors state that “most so-called misinformation originates from strategic efforts” and that “misinformation research predominantly examines belief in disinformation” anyways. However, I am not convinced that the mere quantity of phenomena should guide definitions of our primary research concepts. Should certain emotions not be studied under the term emotions because they are shown less often by humans than other emotions or because researchers have not focused on them so much?

b) Adding to the former, I do not even think that most misinformation originates from strategic efforts. The authors do not provide any references or evidence that that is actually the case, and I can think of several instances where misinformation as a result of error is highly important and potentially common. For example, young parents often report that HPV vaccination might be the cause for their children to become sexually active and health-care workers are often reporting that they do not take influenza vaccination because the vaccine can cause the disease. These errors are often the cause of fallacies and reasoning errors such as correlation is not causation and there is no need to think of a broader disinformation campaign here. In fact, a lot of misinformation can be caused by attitude roots such as moral concerns, fears or unwarranted beliefs (Fasce et al., 2023). Not including reasoning errors of concerned mothers and health care workers on health decision making and the misleading messages they spread (e.g., on social media) as part of misinformation research, excludes a broad and highly relevant part of studies and real life issues.

Fasce, A., Schmid, P., Holford, D. L., Bates, L., Gurevych, I., & Lewandowsky, S. (2023). A taxonomy of anti-vaccination arguments from a systematic literature review and text modelling. *Nature human behaviour*, 7(9), 1462-1480.

b) Moreover, what about more fundamental misinformation research that has often used errors in reports about, for example, a fire in a building to study the impact of phenomena such as familiarity, or continued influence (e.g., Susmann & Wegener, 2022)? According to the new proposed definition these highly relevant studies would not be even considered misinformation research anymore. By doing that the community would take away some of their more theory-based and context-unspecific research findings.

Susmann, M. W., & Wegener, D. T. (2022). The role of discomfort in the continued influence effect of misinformation. *Memory & Cognition*, 50(2), 435-448.

The authors also state that “behavioral science must adopt a broader research agenda focused on understanding the drivers of disinformation behavior and identifying effective interventions to reduce its prevalence.” Some work has been done on attitude roots that addresses this agenda, for example, Hornsey and Fielding 2017.

Hornsey, M. J., & Fielding, K. S. (2017). Attitude roots and Jiu Jitsu persuasion: Understanding and overcoming the motivated rejection of science. *American psychologist*, 72(5), 459.

I sometimes have a hard time following the reasoning. For example, the authors state: “This evolutionary perspective highlights why debunking alone is not optimal: falsehoods are often socially useful to those who spread them. Just as some individuals use gossip to solidify alliances or undermine rivals, groups deploy disinformation for strategic advantage. Thus, interventions focused solely on changing individual minds without addressing the strategic value of disinformation for those who spread it are fundamentally incomplete.” What does fundamentally incomplete mean? Can debunkings become more effective by increasing the persuasive knowledge of the receiver? Because that is exactly what a lot of inoculation interventions already aim to do (e.g., Ziemer et al., 2024). Moreover, research shows that debunking approaches are often successful in reducing the impact of misinformation. What exactly does the evolutionary perspective add there?

Ziemer, C. T., Schmid, P. M., Betsch, C., & Rothmund, T. (2024). Identity is key, but Inoculation helps—how to empower Germans of Russian descent against pro-Kremlin disinformation.

Another thing I struggle with is how everything we would do to counter misinformation would rely on the assumption we are tackling “aggression”. It might start in some cases as aggression but quickly turns into talking points that are just repeated because people actually start to believe them. How do you tackle “aggression” that is not meant to be aggression but a mere expression of beliefs? (other than with all the interventions that have been shown to be effective so far) In fact, extreme science deniers are often not the actual problem but the amount of people they can convince with their messages who then start to repeat/believe/act upon those statements. Framing those as aggressors rather than mislead individuals seems to

increase rather than decreases polarization and also miss the point.

As a final remark, I think some ideas are actually interesting in this piece (e.g., studying systematic interventions that address actual deliberate deniers) but I see no need and actually some harm in concluding that we need to redefine misinformation for this.

Version 1:

Decision Letter:

** Please ensure you delete the link to your author homepage in this e-mail if you wish to forward it to your co-authors **

Dear Dr Adam-Troian,

Your Comment titled "Fake News, Real War: Reconceptualizing Misinformation as Aggressive Behavior." has now been editorially reviewed, and I am delighted to say that we are happy, in principle, to publish it in Communications Psychology.

If the revised paper is in Communications Psychology format, in an accessible style, and of appropriate length, we shall accept it for publication immediately. I have attached an edited version of your manuscript, and ask you to attend to each comment in detail.

EDITORIAL REQUESTS:

* Please review the changes in the attached copy of your manuscript, which has been edited for style, and address the comments and queries I have added. If using Word, please use the 'track changes' feature to make the process of accepting your manuscript more efficient.

* Communications Psychology uses a transparent peer review system. On author request, confidential information and data can be removed from the published reviewer reports and rebuttal letters prior to publication. If you are concerned about the release of confidential data, please let us know specifically what information you would like to have removed. Please note that we cannot incorporate redactions for any other reasons.

*If you have not done so already, please alert me to any related manuscripts from your group that are under consideration or in press at other journals, or are being written up for submission to other journals (see www.nature.com/authors/editorial_policies/duplicate.html for details).

Communications Psychology is a fully open access journal. Articles are made freely accessible on publication. For further information about article processing charges, open access funding, and advice and support from Nature Research, please visit <https://www.nature.com/commpsychol/open-access>

At acceptance, you will be provided with instructions for completing the open access licence agreement on behalf of all authors. This grants us the necessary permissions to publish your paper. Additionally, you will be asked to declare that all

required third party permissions have been obtained, and to provide billing information in order to pay the article-processing charge (APC).

Please note that your paper cannot be sent for typesetting to our production team until we have received this information; **therefore, please ensure that you have this ready when submitting the final version of your manuscript.**

ORCID

Communications Psychology is committed to improving transparency in authorship. As part of our efforts in this direction, we are now requesting that all authors identified as 'corresponding author' create and link their Open Researcher and Contributor Identifier (ORCID) with their account on the Manuscript Tracking System (MTS) prior to acceptance. ORCID helps the scientific community achieve unambiguous attribution of all scholarly contributions. For more information please visit <http://www.springernature.com/orcid>

For all corresponding authors listed on the manuscript, please follow the instructions in the link below to link your ORCID to your account on our MTS before submitting the final version of the manuscript. If you do not yet have an ORCID you will be able to create one in minutes.

IMPORTANT: All authors identified as 'corresponding author' on the manuscript must follow these instructions. Non-corresponding authors do not have to link their ORCIDs but are encouraged to do so. Please note that it will not be possible to add/modify ORCIDs at proof. Thus, if they wish to have their ORCID added to the paper they must also follow the above procedure prior to acceptance.

To support ORCID's aims, we only allow a single ORCID identifier to be attached to one account. If you have any issues attaching an ORCID identifier to your MTS account, please contact the [Platform Support Helpdesk](http://platformsupport.nature.com/).

Link Redacted

We hope to hear from you within two weeks; please let us know if the process may take longer.

Best regards,

Jennifer Bellingtier

Jennifer Bellingtier, PhD
Senior Editor
Communications Psychology

Dear Dr. Schiffer and Dr. Bellingtier,

First of all, I wish to sincerely thank you and the reviewers again for your invaluable input and time spent on this manuscript. As per your suggestions, I am submitting a revised version of the paper. This version contains edits in response to both editorial and reviewer's feedback. For ease of reading, edits are highlighted in yellow throughout the manuscript.

#1 Editor

The present version of this Comment includes a strong and important call to action: focusing on the sources, or perpetrators of misinformation (and considering the spread of misinformation as relational aggression). As you lay out in the beginning and later again, this is conceptually equivalent to focusing on the part of misinformation that is the consequence of disinformation, or put differently, focusing on the perpetrators and mechanisms of disinformation. To make this argument, it is not necessary to deny the contributions that misinformation research (the "cognitive paradigm") has made, nor to argue that all misinformation is the result of deliberate nefarious action (see Reviewer #2). The way the piece is currently written, it needlessly antagonizes readers who will feel that misinformation research that focuses on the recipients is given short shrift and that your proposal offers as an antithesis a similarly reductionist view when a synthesis of both streams would move the field much farther forward. In other words, misinformation research has offered insights into what makes people vulnerable to misinformation and what protects from misinformation spread regardless of whether it's maliciously spread misinformation (i.e. outcome of disinformation) or not. As you explain, it is equally or even more important to focus on actors and on the type of misinformation that is the outcome of disinformation, but that's not to invalidate the existing science and it can be achieved within the nomenclature (misinformation vs disinformation) that you confirm to be largely useful. Although controversy and what Reviewer #1 calls "pithy" arguments can contribute to the success of a piece, it is equally important to get readers on board and avoid statements that they would consider strawman arguments regarding past contributions.

I agree that the tone of some sentences and paragraphs might have been perceived as unnecessarily antagonistic, which was not intended. As you will throughout the manuscript, edits were made to soften the argument and to present the introduction of the proposed "relation aggression" paradigm as complementary (not mutually exclusive) to the prevailing "cognitive paradigm".

Reviewer

#1

1. This commentary argues that misinformation is frequently actually disinformation (i.e., created and spread with intent), and that therefore the conventional research focus on recipients (what the author calls the "cognitive paradigm") is misplaced. Instead, focus should be on the actors that disseminate disinformation with the intent to cause harm. On this view, misinformation is best understood as "a form of aggression that seeks to damage others through social manipulation", which renders interventions that focus on individual

minds “fundamentally incomplete.” An alternative research agenda should (1) classify disinformation tactics, (2) develop predictive models of disinformation behaviors, and (3) design systemic interventions.

Turning to evaluation, I am in two minds about this paper: On the one hand, I agree with almost everything that is said, and I found the pithy and straightforward argumentation quite refreshing. On the other hand, I am not sure whether the “bite size” of this paper is appropriate for the journal. This reads more like a brief opinion piece than an academic article that is extensively supported by citation and evidence. The invitation to review unfortunately did not indicate what type of paper this is, but I checked the submission types (<https://www.nature.com/commspsychol/submit/content-types#comment>) and if this is intended as a Comment, then I would definitely support publication.

I wished to sincerely thank the reviewer for their appreciation of the manuscript and review. As the reviewer suggests, the piece is indeed intended as a comment.

2. 16 A citation in support of this distinction might be helpful here.

The following citation has been added to support the use of biological analogies by misinformation researchers:

Van der Linden, S., Maibach, E., Cook, J., Leiserowitz, A., & Lewandowsky, S. (2017). Inoculating against misinformation. *Science*, 358(6367), 1141-1142.

3. 114 According to the journal, “Readers are alerted that the reliability of data and conclusions presented in this manuscript is currently in question. Appropriate editorial action will be taken once this matter is resolved.” This may not be the best source to cite until this issue is resolved?

Following the reviewer’s observation, we modified this example: “For example, recent research focusing on corporate misinformation indicates that legal consequences—such as imposing severe penalties for spreading misinformation online—can indeed have deterrent effects” and used the reference below:

Crowley, R.M., Lou, Y., Tan, S. T., & Zhang, L. (2025). Misinformation regulations: Early evidence on corporate social media strategy. *Review of Accounting Studies*, 1.

Reviewer #2

In the manuscript “Fake News, Real War: Reconceptualizing Misinformation as Aggressive Behavior.” The authors argue that there is a need to reconceptualize our understanding of “misinformation” and that an aggressive behavior perspective offers a solution. After reading the arguments I remain unconvinced. I explain my concerns in more detail below.

I wished to thank the reviewer for their feedback on the manuscript and review. I hope my response to their observations will address their concerns and increase their confidence in the validity of the arguments developed in this piece.

1. At several occasions the authors argue that “Misinformation (...) is the downstream effect of deliberate disinformation campaigns.” Or that “Misinformation stems from deliberate acts of proactive intergroup aggression,” or that “Misinformation can be defined as the successful outcome of disinformation behaviors.” With this framing the authors exclude false information that is the result of errors from the definition of misinformation. I can only think of two settings why we, as a community, should deviate from earlier definitions and follow that reasoning (e.g.): a) the old definitions are too broad b) the old definitions are too narrow.

I agree with the reviewer and the starting point of the reflection fleshed out in the paper is precisely the realization that the current definition of misinformation may be too broad, as explained in the beginning of section 2:

“While the distinction between misinformation and disinformation is useful, it remains problematic if misinformation is simply viewed as false information. In this case, misinformation research becomes indistinguishable from the broader study of belief, overlapping substantially with core areas of psychology.”

2. Common definitions of misinformation (e.g., Vraga & Bode, 2020) do include disinformation and thus also deliberate disinformation campaigns and deliberate acts of spreading false or misleading content. Following this, I do not see how this situation causes the need for reconceptualization here and the authors also introduce the known difference between misinfo and disinfo. That being said, it could be interesting to study the effectiveness of interventions such as sanctions against malevolent actors but this interesting research question does not imply the need to redefine a concept that allows to do just that, that is, study interventions against disinformation.

Here too I align with the reviewer’s view. I propose that disinformation is not just misinformation plus intent, but that we can better understand misinformation as a consequence of disinformation. And the theory that allows us to make this connection is relational aggression. In other words, I do not support the idea that all misinformation research is ill-defined and should be discarded. Just that there may be another – complementary – way to look at the issue. To make this clearer, I also added in section 2 that:

“Our argument does not deny the importance of research focusing on recipients or on misinformation that is not the result of malicious actors. Rather, it highlights the need to shift attention toward the intentional actors behind disinformation campaigns, an angle that has been overlooked but is crucial for addressing the root causes of the problem.”

3. That leaves us with the recent definitions being potentially too broad. In line with this, the authors state that “most so-called misinformation originates from strategic efforts” and that “misinformation research predominantly examines belief in disinformation” anyways. However, I am not convinced that the mere quantity of phenomena should guide definitions of our primary research concepts. Should certain emotions not be studied under the term emotions because they are shown less often by humans than other emotions or because researchers have not focused on them so much? Adding to the former, I do not even think

that most misinformation originates from strategic efforts. The authors do not provide any references or evidence that that is actually the case, and I can think of several instances where misinformation as a result of error is highly important and potentially common. For example, young parents often report that HPV vaccination might be the cause for their children to become sexually active and health-care workers are often reporting that they do not take influenza vaccination because the vaccine can cause the disease. These errors are often the cause of fallacies and reasoning errors such as correlation is not causation and there is no need to think of a broader disinformation campaign here. In fact, a lot of misinformation can be caused by attitude roots such as moral concerns, fears or unwarranted beliefs (Fasce et al., 2023). Not including reasoning errors of concerned mothers and health care workers on health decision making and the misleading messages they spread (e.g., on social media) as part of misinformation research, excludes a broad and highly relevant part of studies and real life issues.

I agree that the quantity of phenomena should not be the sole criterion for defining research concepts. My intention was to highlight the importance of focusing on misinformation that results from deliberate disinformation campaigns, not to exclude other forms of misinformation, such as errors or misconceptions. I recognize that misinformation can arise from attitude roots, moral concerns, and reasoning errors, as you rightly pointed out.

Regarding your examples of HPV and influenza vaccination misinformation, we agree that these are relevant for misinformation research. Our focus on disinformation as relational aggression is meant to complement existing research, not to exclude these important areas.

The statement about "most misinformation originating from strategic efforts" could have been clearer. I have now included references to studies which show how prevalent strategic disinformation campaigns are to better support this argument (e.g., the issue of "superspreaders" and the political weaponization of misinformation; Altay, 2022; Tornberg & Chueri, 2025).

Altay, S. (2022). How effective are interventions against misinformation?

Törnberg, P., & Chueri, J. (2025). When Do Parties Lie? Misinformation and Radical-Right Populism Across 26 Countries. *The International Journal of Press/Politics*, 19401612241311886.

4. Moreover, what about more fundamental misinformation research that has often used errors in reports about, for example, a fire in a building to study the impact of phenomena such as familiarity, or continued influence (e.g., Susmann & Wegener, 2022)? According to the new proposed definition these highly relevant studies would not be even considered misinformation research anymore. By doing that the community would take away some of their more theory-based and context-unspecific research findings.

The aim of the comment is to complement these existing research efforts by highlighting the strategic and intentional aspects of misinformation. I believe that both approaches are necessary for a comprehensive understanding of misinformation. Objectively, I fail to see how proposing a complementary perspective could deter colleagues from conducting and publishing such fundamental studies, as they are essential to the field.

5. The authors also state that “behavioral science must adopt a broader research agenda focused on understanding the drivers of disinformation behavior and identifying effective interventions to reduce its prevalence.” Some work has been done on attitude roots that addresses this agenda, for example, Hornsey and Fielding 2017.

This relevant piece is now cited.

6. I sometimes have a hard time following the reasoning. For example, the authors state: “This evolutionary perspective highlights why debunking alone is not optimal: falsehoods are often socially useful to those who spread them. Just as some individuals use gossip to solidify alliances or undermine rivals, groups deploy disinformation for strategic advantage. Thus, interventions focused solely on changing individual minds without addressing the strategic value of disinformation for those who spread it are fundamentally incomplete.” What does fundamentally incomplete mean? Can debunkings become more effective by increasing the persuasive knowledge of the receiver? Because that is exactly what a lot of inoculation interventions already aim to do (e.g., Ziemer et al., 2024). Moreover, research shows that debunking approaches are often successful in reducing the impact of misinformation. What exactly does the evolutionary perspective add there?

I appreciate the reviewer’s concern regarding the clarity of my reasoning. When I state that interventions focused solely on changing individual minds are "fundamentally incomplete," I mean that they do not address the strategic and intentional aspects of disinformation. While debunking and inoculation interventions are indeed valuable and often effective in reducing the impact of misinformation, they primarily target the recipients rather than the actors who spread disinformation.

The evolutionary perspective adds to this by highlighting that falsehoods are often socially useful to those who spread them, much like gossip. This means that interventions must also address the strategic value of disinformation for those who deploy it. According to this view, spreaders are necessarily motivated actors because spreading gossip is not a “passive” behavior reflecting naïve beliefs about a topic— it most often a strategic, reputation enhancing behavior aimed at denigrating a target (group or individual) associated with the topic. In fact, evolutionary informed research shows how few individuals share fake news because it generally damages their reputation (Altay et al., 2022). Spreaders are thus those who see reputational benefits in it. In terms of application, this adds name-and-shame type of responses in the toolbox of viable routes to deter disinformation (see Unver & Arhan, 2023). This point is now discussed in the relevant section of the paper.

Altay, S., Hacquin, A. S., & Mercier, H. (2022). Why do so few people share fake news? It hurts their reputation. *New Media & Society*, 24(6), 1303-1324.

Unver, H. A., & Arhan, S. E. (2023). The Strategic Logic of Digital Disinformation: Offence, Defence and Deterrence in Information Warfare. In *Routledge Handbook of Disinformation and National Security* (pp. 192-207). Routledge.

7. Another thing I struggle with is how everything we would do to counter misinformation would rely on the assumption we are tackling “aggression”. It might start in some cases as

aggression but quickly turns into talking points that are just repeated because people actually start to believe them. How do you tackle “aggression” that is not meant to be aggression but a mere expression of beliefs? (other than with all the interventions that have been shown to be effective so far) In fact, extreme science deniers are often not the actual problem but the amount of people they can convince with their messages who then start to repeat/belief/act upon those statements. Framing those as aggressors rather than mislead individuals seems to increase rather than decreases polarization and also miss the point.

Thank you for raising this important point. I understand your concern about how my approach to countering misinformation might be perceived as relying solely on the assumption that we are tackling "aggression."

It is important to clarify that the focus on disinformation as relational aggression does not imply that all expressions of misinformation are inherently aggressive. I recognize that misinformation can evolve from strategic aggression into widely held beliefs, repeated by individuals who genuinely believe them. The argument is not to label all individuals who spread misinformation as aggressors, but rather to target the intentional actors who initiate and propagate disinformation campaigns.

Moreover, looping back to the reviewer's point #3, it seems that empirically, most of misinformation is the product of a few superspreaders (Altay, 2022). If social contagion induction effects apply to misinformation incapacitating these individuals through social network targeting strategies could lead to decreases in misinformation prevalence equivalent to interventions on the whole population (see Airoidi & Christakis, 2024).

Airoidi, E. M., & Christakis, N. A. (2024). Induction of social contagion for diverse outcomes in structured experiments in isolated villages. *Science*, 384(6695), eadi5147.

Altay, S. (2022). How effective are interventions against misinformation?